# Aging in the Right Place for Older Adults Experiencing Housing Insecurity: An Environmental Assessment of Temporary Housing Program

**DOI:** 10.3390/ijerph192214857

**Published:** 2022-11-11

**Authors:** Atiya Mahmood, Rachelle Patille, Emily Lam, Diana Juanita Mora, Shreemouna Gurung, Gracen Bookmyer, Rachel Weldrick, Habib Chaudhury, Sarah L. Canham

**Affiliations:** 1Department of Gerontology, Simon Fraser University, 515 West Hastings, Suite 2800, Vancouver, BC V6B 5K3, Canada; amahmood@sfu.ca (A.M.); shreemouna_gurung@sfu.ca (S.G.); rachel_weldrick@sfu.ca (R.W.); habib_chaudhury@sfu.ca (H.C.); 2Faculty of Health Sciences, Simon Fraser University, 8888 University Dr, Burnaby, BC V5A 1S6, Canada; ehl8@sfu.ca (E.L.); diana_mora@sfu.ca (D.J.M.); 3College of Social Work, University of Utah, 395 S 1500 E, Salt Lake City, UT 84112, USA; sarah.canham@utah.edu; 4College of Architecture and Planning, 375 1530 E, University of Utah, Salt Lake City, UT 84112, USA

**Keywords:** aging in the right place, older adults, built environment, housing insecurity, homelessness

## Abstract

Research on programs offering senior-specific housing supports and enabling “aging in the right place” (AIRP) for “older persons with experiences of homelessness” (OPEH) is limited. This paper presents an environmental assessment of a “transitional housing program” (THP) in Metro Vancouver, Canada, for OPEH to AIRP. Data were collected using Aging in the Right Place Environmental (AIRP-ENV) and Secondary Observation (AIRP-ENV-SO) audit tools designed to evaluate multi-unit housing for OPEH. The 241-item AIRP-ENV tool was used to assess the built environmental features of four multi-unit buildings of the THP. The AIRP-ENV-SO tool was used to collect contextual data on the function, safety, and land use of the surrounding neighborhood. Findings identified built environment and urban design features that support THP residents’ safety, security, accessibility, functionality, social activity, autonomy, and identity. The THP buildings were rated ‘*Good*’ for accessibility, functionality, autonomy and identity, while ‘*Satisfactory*’ or ‘*Poor*’ for safety, security, and social activity. Findings point to the built environmental features (e.g., size and layout of spaces) required in the THP to create opportunities for increased social engagement among residents and enhanced safety and security. The AIRP-ENV and AIRP-ENV-SO audit tools can help inform programs across the housing continuum to develop supportive built environments that promote AIRP for OPEH.

## 1. Introduction

The global aging population is increasing due to declining fertility and increasing life expectancy influenced by economic and social growth. By 2030, the world population of those aged 60 and over is expected to grow by 56%, increasing from 901 million to 1.4 billion [1]. Related to this trend of population aging is the increase in homelessness among older adults in Canada [2]. In Metro Vancouver, Canada, wide-ranging factors, such as the high cost of living, low income, limited opportunity to rejoin the workforce, challenging health conditions, and increased social isolation, have increased experiences of housing insecurity for older adults [2]. Older adults, defined as those aged 55 and older, made up 24% of the homeless population in Metro Vancouver in 2020, which is an increase from 22% in 2017 [3]. The Canadian Observatory on Homelessness defined homelessness as “the situation of an individual, family or community without stable, safe, permanent, appropriate housing, or the immediate prospect, means and ability of acquiring it” [4].

The need to address housing insecurity and challenges of care placement for older adults, among others, has prioritized aging in place (AIP) in public policy, program development, and service delivery. AIP “broadly refers to the notion of aging in one’s home and community as long as possible and to delaying any potential relocation to a long-term care setting” [5] (p. 235). However, AIP for older persons with experiences of homelessness (OPEH) can often be infeasible or challenging. OPEH who are staying in emergency shelters or transitional housing do not have the privilege of a permanent and stable home to AIP [6]. 

Housing instability can negatively impact the health and well-being of older adults [7,8,9]. Research suggests that OPEH have mental and physical health characteristics that are similar to non-homeless individuals who are 10 to 20 years older [10,11]. Additionally, OPEH have higher rates of early mortality and longer shelter stays than younger homeless people [10]. Although recognized as an important issue in the housing sector, AIP cannot be viewed as a universal policy for all older adults as it does not apply to certain subpopulations, such as OPEH [12]. Instead, aging in the right place (AIRP) recognizes that secure and optimal housing should support an individual’s unique vulnerabilities and lifestyles [13]. AIRP acknowledges that OPEH may encounter a variety of challenges in accessing financial, social, and health resources that could contribute to negative health consequences.

While policies that support AIRP have the potential to improve the lives of the homeless population, there is a gap in the literature on topics related to OPEH [12]. Though there are programs and policy initiatives aiming to reduce homelessness across Canada, limited attention is directed toward OPEH, their unique challenges, and evaluations of housing interventions tailored to OPEH [12]. Extensive literature highlights the benefits of AIRP and recognizes the desire of older adults to age in their communities [14,15,16,17]; however, this literature largely overlooks the significance of AIRP for OPEH [14].

Considering the increase in the aging population and OPEH, there is a need for conceptual understanding and empirical evidence on how this population can more effectively AIRP. To address this gap, Canham and co-authors (2022) developed a conceptual framework consisting of key indicators of AIRP for OPEH: (1) Built environment of the housing unit and surrounding neighborhood; (2) Off-site and on-site health and social services and resources; (3) Social integration; (4) Stability and affordability of place; (5) Emotional place attachment; and (6) Broader political and economic contexts [18]. The first indicator, “built environment of the housing unit and surrounding neighborhood” is of particular interest, as an unsupportive or challenging environmental context can have detrimental and cascading effects on the experience of OPEH [19]. The built environment is generally understood as “the buildings and landscapes people inhabit… [including] both indoor and outdoor spaces at a variety of scales” [1] (p. 182). Suitable, safe, and accessible built environments are foundational to AIRP; thus, evaluation of the environmental context is key to identifying, modifying, and guiding new housing developments and support services that can support AIRP for OPEH [1].

Lawton and Nahemow (1973) first conceptualized aging and environment as involving a Person-Environment (P-E) dynamic interaction to describe the relationship between aging individuals and their physical environment [20]. Lawton and Nahemow (1973) argued that older individuals are especially affected by the P-E dynamic as the P-E interaction isconceptualized as a core element of older adults’ stability and quality of life; however, not until recently has gerontological research highlighted the significance of physical environment in AIP; [21,22]. The built environment, including transportation, outdoor areas, and housing, impacts one’s mental health and social connectivity [19]. For example, appropriately planned outdoor areas can provide opportunities for social interactions and activities and, in turn, decrease loneliness and isolation among older adults, while inadequate or unsupportive built environments can have an opposite effect [19]. Neighborhoods also play a significant role in fostering autonomy, independence, and social interactions for older adults [1,23]. Recent work by Wang et al. (2021) has identified that later life conditions are influenced by the environment where individuals lived in earlier life stages, implicating the life course impacts of one’s environment on their health outcomes [24].

Given the importance of the built environment on older adults’ ability to AIRP, research on environmental factors that contribute to AIRP for OPEH is needed. Environmental audit tools offer an effective means to evaluate the environment both objectively and subjectively and identify factors that may impact AIRP [1]. Thus, environmental audits of housing sites for OPEH are needed to understand the facilitators and barriers to housing stability and access that can support or hinder AIRP. Data collected through environmental audits can guide interventions and actions that support the development of age-friendly communities and inclusive housing [25]. This, in turn, can facilitate the development of proactive environmental approaches that support healthy and active aging for diverse groups of older adults, including OPEH.

## 2. Study Objective

We conducted environmental audits of the built environmental features of a temporary housing program (THP) for older adults who are homeless or at risk of being homeless in Metro Vancouver, Canada. The THP provides a temporary furnished private apartment for a period of three to six months while THP clients seek permanent housing with the support of a housing worker. The THP supports clients to live in independent market rental apartment buildings where clients are unsupervised and unattended. As no prior research has conducted environmental audits of housing units for OPEH, the goal of our study was to audit the environmental features of THP units based on factors that the existing literature has identified as contributing to well-being, safety, and stability in multi-unit housing units for general populations of older adults. Our audit included items that examined the following concepts of the built environment: *Functionality, Accessibility, Safety and Security, Social Activity, and Autonomy and Identity*. These concepts are identified within the AIRP framework [12] as built environment features that can facilitate or create barriers to AIRP for OPEH. This environmental audit is a part of a larger multi-year and multi-city mixed-method study that is focused on discovering AIRP factors across different types of housing (referred to as “promising practices” in the study) that are targeted for OPEH across the housing continuum. The promising practices range from shelters to residential care. Promising practices are innovative models of shelter, temporary housing, and other housing supports that hold promise in supporting AIRP for OPEH; however, they have yet to undergo rigorous evaluation [26]. These promising practices are models that have the potential for ‘scaling up’ and ‘scaling out’ into additional contexts with the potential of being elevated to the level of ‘best practice’.

## 3. Methods

The Aging in the Right Place Environment (AIRP-ENV) and the Secondary Observation (AIRP-ENV SO) Audit Tools were developed to evaluate the built environmental quality of (1) permanent supportive housing, (2) supported housing, (3) transitional housing, and (4) emergency shelter settings with health and social supports [12] (see Appendix A)

### 3.1. Audit Tools

The AIRP-ENV tool was adapted from the Physical and Architectural Features Checklist (PAF) and the Multiphasic Environmental Assessment Procedure (MEAP) [27] to assess how the built environment can either facilitate or hinder the ability of OPEH to AIRP. This tool includes 241 items structured in four sections: (A) Exterior of the building (e.g., main entrance, gardens/landscaping, parking area), (B) Communal and shared areas of the interior of the building (e.g., social and recreational spaces, hallways, different amenities, and presence of support services), (C) Residents’ units, and (D) Overall building: rating of the housing’s appearance (e.g., interior room décor and furnishings).

The AIRP-ENV SO tool was developed to assess the neighborhood surrounding the targeted building. This tool was based on the Stakeholders Walkability/Wheelability Audit in Neighborhoods (SWAN) tool [28], which is used to evaluate the walkability and wheelability of neighborhoods by diverse groups of users, including older adults. While the AIRP-ENV tool collects quantitative data on the presence and absence of objective features of the built environment of the housing, the AIRP-ENV SO tool collects qualitative data on the environment of the surrounding neighborhood.

### 3.2. Data Collection

Data were collected at a Temporary Housing Program (THP) that received housing support services through Grandview Heights Services (GHS), a pseudonym for a non-profit senior-serving agency in Metro Vancouver, British Columbia (BC), Canada. Additional pseudonyms are used for the four GHS housing sites reported in this paper. GHS provides local and provincial programs and services to support OPEH to live independently with dignity in a safe, comfortable, and healthy home. OPEH living in the GHS-supported THP live in four multi-unit buildings that we will call Hallow Place, Arbutus Plaza, Meadow Gardens, and Sunset Place. AIRP-ENV and AIRP-ENV SO audits were conducted by two trained researchers (auditors) at each site at the beginning of 2021. These audits were completed while there were still COVID-19 pandemic restrictions in place; thus, access was restricted to certain areas of the buildings, such as staff office space, laundry rooms, and communal kitchens. In one of the housing locations, auditors did not have the opportunity to access a resident’s unit. Thus, for each item of the tool, response options included “Yes”, “No”, “N/A”, and “No Access” (indicating COVID-19 restrictions). Each auditor independently conducted the audit using a paper copy of the tool. Data were collected by two team members for interrater reliability. Our team assigned one of the auditors as the primary data collector and the other as a backup data collector (whose data will mainly be used to run interrater tests). The findings discussed in this paper are based on the data collected by the primary auditor.

### 3.3. Data Aggregation

A digitized version of the tool was developed in SurveyMonkey for data entry, management, and analysis. Once the audit data were digitally compiled, the database was extracted into an Excel file for analysis. The raw data were double-checked for blank entries and to decipher between “N/A” and “No Access” responses. Any confusion or adjustments were validated and confirmed by the researchers who collected the data. Scores for “Yes” (presence of a feature) were denoted as “1” and “No” as “0.” “N/A” and “No Access” responses were excluded from analysis as irrelevant. Data collected with the AIRP-ENV SO tool were transcribed and imported into the qualitative data analysis software NVivo 12 for data analysis [29,30].

### 3.4. Data Analysis

To identify the built environmental facilitators or barriers for OPEH to AIRP, data from Sections A, B, and C of the AIRP-ENV tool were categorized into an analytic framework of higher-level concepts linked to three indicators in the AIRP conceptual framework: (1) built environment of the housing unit and surrounding neighborhood, (3) social integration, and (5) emotional place attachment [18]. The aggregation of individual quantitative items facilitated the creation of the higher-level concepts of *Functionality, Accessibility, Safety and Security, Social Activity, and Autonomy and Identity*. This regrouping of data into five higher-order analytical categories generated overall baseline scores for each category. The category of *Functionality* was linked to 155 items, *Accessibility* to 104, *Safety and Security* to 120, *Social Activity* to 83, and *Autonomy and Identity* to 10 items of the audit tool. Each GHS building was analyzed separately as design and access differed across buildings, resulting in different baseline scores for each conceptual category for each of the four buildings. Baseline scores represented an aggregate of items that were applicable and present for each analytical category.

The AIRP-ENV tool questions were weighted to minimize any possible overrepresentation of certain features. For example, a question regarding “seating” has 5 sub-questions, whereas a question related to “lighting” has 2 sub-questions. If each sub-question within these two features receives a score of 1, then seating as a feature gets a higher score and is shown as worth more in the analysis than lighting. To avoid this type of issue, the sub-questions were weighted, resulting in the main question receiving a score of 1, and any associated sub-questions weighted so that they all added to a score of 1. That is, each individual sub-questions received a decimal place score depending on how many sub-questions were present; for example, 5 sub-questions, each worth 0.2 to add up to a total score of 1; or 2 sub-questions, each worth 0.5 to add up to a score of 1. This resulted in both seating and lighting having equal weight. Further, there were certain questions in the AIRP-ENV tool that were reverse-coded. For example, a positive response to the presence of a certain physical feature (e.g., “are there raised thresholds at the entrance”) indicates it is a barrier to AIRP. This question, if answered “Yes”, would be scored as “1” though it is not a facilitator to AIRP. To avoid this type of issue, questions were reverse scored, as needed, to ensure the total score for each concept only included “Yes” scores.

After weighing and reverse coding, a total score for each concept was calculated. The total score for each concept was then divided by the baseline score for each building to calculate a percentage score for the concept. Percentage scores were organized along a 5-point Likert scale from “*Poor*” to “*Excellent*” (Table 1), indicating the presence rating for different positive environmental features. A percentage score of 20% or less is a *Poor* presence rating, with only a few positive environmental features present, while a rating of 81% or above is an *Excellent* presence rating. This five-point rating scale allows for a graded ranking rather than a binary ‘bad’ versus ‘good’ rating.

Qualitative data from the AIRP-ENV SO audit were analyzed to identify the built environment aspects of the neighborhoods surrounding the THP units. Using NVivo 12, the AIRP-ENV SO audit data were coded line-by-line and subsequently organized into categories. This enabled the development of a codebook of barriers to and facilitators of AIRP for OPEH that was reviewed and approved by all team members.

## 4. Findings

Ratings of all four THP buildings were *Good* (42.1% of the time), *OK* (47.4% of the time), and *Fair* (10.5% of the time) for the concepts of *Functionality* (F), *Accessibility* (A), *Safety and Security* (SS), *Social Activity* (SA), and *Autonomy and Identity* (AI). Specifically, residents’ units were rated *Good* (58.3% of the time), the interiors of the buildings were rated *OK* (56.3% of the time), and the exteriors of the buildings were rated *Excellent* or *Good* (75% of the time). Table 2 provides a breakdown of scores for each site by sections (exterior, interior, and residential unit) for each AIRP concept. Table 3 outlines the overall rating for each building, per AIRP concept. In the following section, the findings are discussed separately for different areas within each building (e.g., exterior and interior of the buildings as well as resident units).

### 4.1. Exterior of the Buildings

The exterior features of the THP buildings were linked to the AIRP conceptual framework indicator “Built Environment and Surrounding Neighbourhood” [18]. The exteriors of all four buildings were rated *Good* or *Excellent* for the concepts of *Functionality* and *Accessibility* due to the presence of built environment features that are compatible with diverse users (e.g., well-maintained landscapes, accessible outdoor benches and seating areas, and available pathways and ramps). *Safety* and *Security* were rated *Good* for Meadow Gardens, *OK* for Arbutus Plaza and Sunset Place, and *Fair* for Hallow Place, suggesting some areas for improvement. *Safety* and *Security* of building exteriors could be improved through the presence of outdoor lighting, visible and well-lit parking, outdoor security cameras, and sheltered seating to combat weather challenges. *Social Activity* ratings were higher; however, to improve this AIRP concept at Arbutus Plaza and Meadow Gardens, *Social Activity* in outdoor spaces could be facilitated through the presence of communal patios, walking areas, additional seating, and gardens. In addition, implementing features such as storage space for personal vehicles (e.g., bicycles, scooters) could contribute to AIRP.

### 4.2. Interior of the Buildings

For the interior environment of the THP buildings, *Functionality* was rated *OK* for three of the four buildings, with the fourth scoring *Good*. For the concept of *Accessibility*, Hallow Place and Arbutus Plaza scored *Good,* whereas Meadow Garden and Sunset Place scored *OK*. We identified a need for adequate lighting, functioning elevators, well-maintained stairs and fire escape, indoor ramps, and sufficient and functioning light fixtures in hallways, indoor lounges, and common areas. Conversely, some supportive features included functional and accessible laundry facilities (where at least two functioning washers and two dryers are present), adequate laundry space and layout (i.e., including location, sink, seating area, bulletin board, clock, counter space for folding), and natural light. *Safety and Security* ratings ranged from *Good* at Sunset Place to *Fair* at Meadow Gardens. Features found to promote *Safety and Security* included the presence of natural light in communal spaces (e.g., dining area, lounge, hallway, entrance) and accessible information signs and systems (e.g., speaker systems, operation hours of communal spaces). The presence of security functions (e.g., camera security system in the lobby and reception area), adequate posting of COVID-19 protocols (e.g., visible signage in different languages), emergency protocols (e.g., signposts outlining emergency contacts and procedures), and accessibility features (e.g., ramps and handrails) would improve *Safety and Security*.

Compared to the other AIRP concepts, *Social Activity* was rated poorer at all four THP buildings. Though the low scores may have been impacted by COVID-19 restrictions, GHS could explore ways to foster *Social Activity* through the design and layout of space and furniture (e.g., multi-purpose rooms, lounges for recreational activities, and furniture that seat two or more people), as well as personal aesthetics (e.g., artwork and other decors that promote engagements among like-minded residents) for residents.

### 4.3. Residents’ Units

Resident unit data were collected for three of the four buildings (Hallow Place, Arbutus Plaza, and Sunset Place). As illustrated in Table 2, these buildings are rated *Good* for *Functionality*, suggesting that built environmental features, such as both natural and artificial light, were present and that light fixtures were functioning in residents’ units. *Accessibility* and *Safety and Security* ratings varied from *Fair* to *Good*, indicating that some features that foster AIRP are present, whereas others are missing. In the *Accessibility* category, the size of the bathroom and hallways in residents’ units were incompatible with the use of assistive devices (e.g., wheelchairs). However, the presence of grab bars were positive features of *Accessibility* and *Safety and Security* in the units [31]. However, *Accessibility* and *Safety and Security* in residents’ units could be improved through modifications that follow universal design principles, such as lower kitchen countertops, wider unit entrances, and adequate bathroom size and layout [32,33,34,35]. Some of these modifications may be possible if funding is available; however, this may be difficult to attain due to limited funding availability for non-profit housing providers as well as design restrictions. Implementing safety measures, including anti-slip surfaces, sprinkler systems, carbon monoxide detectors, fire extinguishers, adequate lighting, and hand railings, could increase the *Safety and Security* ratings of residents’ units.

The AIRP concept of *Autonomy and Identity* scored *Good* in two of the three THP buildings, whereas Arbutus Plaza scored an *OK* rating. This finding highlights the opportunity to enhance *aAutonomy and iIdentity* by changing design and organizational policies. For example, the provision of adequate space and lockable storage for personal belongings, as well as facilitating unit personalization (e.g., wall décors and furnishing) to create home-like environments can contribute to one’s sense of *Autonomy and Identity* by increasing feelings of feeling known, valued, and belonging [36].

### 4.4. Surrounding Neighbourhood

The surrounding neighborhoods of all four THP buildings included built environmental features that both positively and negatively impacted OPEH to AIRP. The neighborhood-built environmental features found to facilitate AIRP included accessible sidewalks and building entrances, traffic calming features, adequate outdoor space and lighting, and the presence of public transportation. Sidewalks and building entrances were rated *Accessible* and, in turn, *Safe* if attributes such as walkability, wheelability, cleanliness, curb cuts, and paved surfaces were present. Traffic calming features include the presence of speed limit signs, stop signs, and traffic lights, as well as crosswalk opportunities for pedestrians. These features, along with slow traffic flow and low traffic noise, quiet residential areas, and a mix of a low and high volume of pedestrians, promote *Safety and Security*, as well as *Social Activity*. The presence of outdoor spaces (e.g., benches and seating areas, parks, walking trails), lighting (e.g., streetlamp, well-lit areas), public amenities (e.g., grocery and drug stores, swimming pool), and public transportation (e.g., transit stops) also contribute to *Functionality*, *Safety and Security*, *Accessibility*, and *Social Activity*. It should be noted that well-maintained areas and the presence of nature are “associated with residents’ increased sense of personal safety”, whereas the presence of “abandoned properties, graffiti, and trash dumping can evoke feelings of fear” and, therefore, generate a perception of an unsafe and insecure environment [37] (p. 804 and 802).

We identified the following barriers in the surrounding neighborhood of the THP buildings: lack of curb cuts and sidewalks, outdoor lights, crosswalks, speed bumps and other traffic circles, traffic lights, grocery stores nearby, bus stops and shelters, and public spaces. The presence of cracked and uneven sidewalks, steep uphill, heavy traffic flow, high traffic speed and noise, and garbage were other barriers noted in the neighborhoods.

## 5. Discussion

The AIRP-ENV and AIRP-ENV SO tools can be used to assess the built environmental features of shelter and housing units, buildings, and neighborhoods for OPEH. Adaptation and development of these built environment audit tools for housing that serves older adults at risk of homelessness or experiencing homelessness is a new contribution to the field. In this article, we demonstrated that data collected through the AIRP-ENV tool can contribute to a balanced evaluation of built environmental facilitators and barriers within THP sites for OPEH. It contributes to determining whether or not these sites support AIRP for OPEH through focused audit data collection across the concepts of *Functionality*, *Accessibility*, *Safety and Security*, *Social Activity*, and *Autonomy and Identity*. Framing and grouping individual responses under five analytical concepts strengthen the findings as these concepts are in turnlinked to key indicators within the AIRP conceptual framework developed by Canham et al. (2022). This type of built environment data reduces the gap that currently exists in the literature on the role environmental factors play in the AIRP of OPEH [12,18].

Based on findings from our AIRP-ENV audit, a recommendation to improve the ability of THP residents to AIRP includes implementing universal design principles to create more accessible features in the residential unit to support OPEH of different functioning abilities, mobility limitations, and cognitive impairments [32,33,34,35]. For example, implementing lower kitchen countertops, wider unit entrances, and bathroom design layouts can support residents in utilizing the space more effectively. Built environmental features that would further support OPEH to AIRP at the THP include the provision of carbon monoxide detectors, anti-slip surfaces, and hand railings that can increase elements of *Safety and Security*. Based on the findings of the AIRP-ENV audit, implementing these features are imperative, as the concept of *Safety and Security* ranked between *Fair* and *Good* in the residential units.

Residential units should provide opportunities for personalization to foster a home-like environment for OPEH and increase feelings of *Autonomy and Identity*. Older adults can exhibit a strong sense of ‘insideness’, which is understood as one’s attachment to their place of residence [38]. Chaudhury and Oswald (2019) point out that developmental outcomes of autonomy and identity are influenced by the interaction between a person and an environment. Therefore, providing appropriate physical space to display personally meaningful belongings in the residential units can create a home-like environment where OPEH can develop a sense of place attachment, which in turn, has the potential to enhance their *Autonomy and Identity* [21,38,39].

To increase *Functionality* of the interior building, adequate lighting, functioning elevators, indoor ramps, and well-maintained stairs must be present. Features that support *Safety and Security* include natural lighting in communal spaces, accessible signage, and information, as well as having security functions throughout the building. Ensuring adequate COVID-19 and emergency protocols with signage were also considered important aspects for increasing *Safety and Security* for OPEH to AIRP within the context of COVID-19. Findings indicated that special attention and consideration should be in place to foster *Social Activity* and create a social environment through built environmental opportunities such as design, layout, and furniture to increase opportunities for *Social Activity* in common rooms/spaces among residents, friends, and loved ones [40]. Creating safe and accessible opportunities for *Social Activity* and engagement indoors can combat isolation and loneliness among OPEH at the THP and support their experiences of AIRP [41,42,43].

The exteriors of the THP buildings were supportive of AIRP, given the presence of accessible seating areas, pathways, and landscape design that were well maintained and supportive for people using assistive devices. These environmental features of access to outdoor spaces and natural environments have been associated with improved physical and mental well-being among older adults [19], which may benefit the overall well-being of OPEH. *Safety and Security* can be improved through outdoor lighting, hand railings, outdoor security systems, and covered seating areas. Improving these features is essential as older people are known to encounter violence on the streets and face higher threats to safety than younger people [44,45]. *Social Activity* ratings could be improved by providing access to communal patios, walking areas, additional seating, and gardening opportunities to support social interactions among residents, features which are often desired by OPEH [46]. The presence of external built environment features that facilitate social activity can reduce experiences of loneliness and isolation among older adults [47].

Findings from the AIRP-ENV SO tool suggest that supportive features of the surrounding neighborhood of the THP include accessible sidewalks and building entrance infrastructure that promotes walkability, wheelability, and cleanliness, all of which promote physical activity among older adults [48]. Additionally, traffic calming features that facilitate slow traffic, low noise, a variability of the volume of pedestrians, and easy access to public transportation were found to be positive neighborhood attributes. Having supportive features, like transportation, available facilitate older adults’ engagement in communities and access to services and supports, which contributes to improved mental health and social connectivity [49,50,51]. Having access to outdoor spaces, walking trails, and public amenities with outdoor seating that are well-maintained and close to nature contributes to a welcoming surrounding neighborhood environment [37]. A supportive surrounding environment of the housing may support OPEH to AIRP in their own homes and communities. However, if these supportive built environmental features are absent or limited in the surrounding community, it can negatively impact the quality of life and independence of OPEH and, in turn, may limit their ability to AIRP [12].

The AIRP-ENV and AIRP-ENV SO tools have the potential to be used to evaluate a variety of shelter and housing types beyond temporary housing. Utilizing these types of audit tools to evaluate the built environment across the housing continuum for OPEH could lead to a better understanding of the built environment and, in turn, contribute to meeting the unique needs of OPEH.

## 6. Limitations

Although this study has multiple benefits, it is important to note its limitations. Since this study was cross-sectional, it was unable to capture environmental changes that occur over time as the tool can only capture the experiences during the time of the audit and, as such, should be interpreted in this way [52,53,54,55]. The authors of the paper recognize that most OPEH residing in THP have lived in and experienced a variety of different physical environments, which may have been problematic and unsafe over their life course. Thus, whether OPEH are able to AIRP in a particular THP may be impacted by their multiple environmental lived experiences (including the experience of living in the current THP).

Capturing the changes that take place in the environment over time is essential for the future of this research as the functional decline of older adults may shape how they experience their surrounding areas, such as their neighborhoods and communities [52]. Future research could employ a longitudinal study design to capture how the person-environment relationship changes over time. Although the tool captures aspects of the built environment, it does not include the voices and lived experiences of OPEH. Including this information would contextualize the facilitators and barriers that OPEH experience within their living environment in relation to AIRP.

Due to the COVID-19 pandemic, research team members were limited in their ability to enter and conduct an audit of certain spaces in the interior of the THP buildings (e.g., some residential units and common spaces). Therefore, it can be argued that the findings pertaining to *Social Activity* may have been impacted by the COVID-19 restrictions as during the time of the audit, residents may have been unable or less inclined to gather socially with public mandates requiring two meters of physical distancing in place and discouraging gathering. Thus, the findings should be interpreted with these limitations in mind.

## 7. Conclusions

The AIRP-ENV and AIRP-ENV SO audit tools can be used to effectively evaluate different types of temporary and permanent supportive housing across the housing continuum for OPEH. By linking the AIRP-ENV and AIRP-ENV SO tools to existing AIRP indicators, this research fills a gap in the literature on the environmental role of the AIRP on OPEH. The findings from the AIRP-ENV tool can support further identification of areas for environmental interventions at housing sites targeted for OPEH. Additionally, the tool can inform planning and design guidelines for housing that support marginalized OPEH who experience a range of individual health and functional statuses. Continued efforts in the THP to improve and adapt environmental features related to the five analytical concepts of *Functionality*, *Accessibility*, *Safety and Security*, *Social Activity*, and *Autonomy and Identity* are warranted. These implementations could improve the well-being and housing stability of OPEH.

## Figures and Tables

**Table 1 ijerph-19-14857-t001:** Five-point AIRP-ENV Presence Rating Scale.

Poor	Fair	OK	Good	Excellent
0–20%	21–40%	41–60%	61–80%	81–100%

**Table 2 ijerph-19-14857-t002:** Sectional scoring of AIRP concepts per THP building.

	Section A: Exterior of the Building				Section B: Communal Areas in the Building				Section C: Residents’ Units				Overall				
Buildings	F	A	SS	SA	F	A	SS	SA	F	A	SS	AI	F	A	SS	SA	AI
Hallow Place	71.6%	100.0%	28.4%	80.8%	57.1%	68.8 %	45.5%	42.1%	68.0%	77.8%	39.4%	62.5%	63.6%	73.0%	41.4%	47.6%	62.5%
Arbutus Plaza	71.1%	100.0%	56.3%	44.8%	63.9%	67.9%	43.5%	50.1%	70.0%	33.3%	42.9%	57.1%	67.1%	64.2%	45.4%	49.4%	57.1%
Meadow Gardens	72.7%	100.0%	80.5%	68.4%	47.1%	54.0%	36.0%	28.5%	No Access	No Access	No Access	No Access	57.3%	59.3%	47.2%	34.4%	No Access
Sunset Place	65.7%	68.8%	45.1%	84.7%	48.4%	56.3%	60.9%	17.9%	71.4%	45.5%	69.4%	71.4%	62.2%	54.6%	60.6%	30.1%	71.4%

Note: F = Functionality, A = Accessibility, SS = Safety and Security, SA = Social Activity, AI = Autonomy and Identity. No Access is due to COVID-19 restrictions.

**Table 3 ijerph-19-14857-t003:** Overall scoring of AIRP concepts per THP building.

Overall Scoring					
Buildings	Functionality	Accessibility	Safety and Security	Social Activity	Autonomy and Identity
Hallow Place	63.6%	73.0%	41.4%	47.6%	62.5%
Arbutus Plaza	67.1%	64.2%	45.4%	49.4%	57.1%
Meadow Gardens	57.3%	59.3%	47.2%	34.4%	No Access
Sunset Place	62.2%	54.6%	60.6%	30.1%	71.4%

## Data Availability

The data are not publicly available due to the ethics protocol of the funding agency.

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
