# Peer review of "Aging in the Right Place for Older Adults Experiencing Housing Insecurity: An Environmental Assessment of Temporary Housing Program"

_ijerph, 2022, doi:10.3390/ijerph192214857_

Round 1

Reviewer 1 Report

This is an intriguing and timely study. The authors leveraged AIRP-ENV and AIRP-ENV-SO audit tools in their study to assess multiunit housing designed for older people experiencing homelessness. The categorized rating was performed to better understand the conditions of these communities. In general, the study is well designed, and the paper is well written, although some parts might need clarification during revision. Here are my specific comments:

Page 2: “Several research and innovative policy initiatives tackling home-lessness are piloted across Canada, but to date little attention is directed towards OPEH and their specific challenges” – It would be helpful to add some more details of the temporary housing program, especially regarding the studied communities. For example, are older people experiencing homelessness still considered “homeless”?

Page 2: “Lawton and Nahemow argued that individuals in their later life are especially more affected by the P-E dynamic (Wahl et al., 2012).” – It should be mentioned that the later life conditions are also partly determined by the environment that individuals lived in across early life stages, as reviewed in a recent article: https://doi.org/10.1038/s43587-021-00140-5.

Page 3: “Emergent research in Environmental Gerontology is demonstrating that the built environment, including transportation, outdoor areas, and housing, has also been linked to positive mental health outcomes and improved social connectivity (Black & Jester, 2020).” Whether positive or negative depends on the characteristics of the built environment. I suggest the authors remove “positive” and “improved”.

Page 4: “These audits were done while there were still COVID-19 Pandemic restrictions in multi-unit housing” – given the conditions during the study, will the pandemic have an impact on the results? (Especially the safety and security and social activity categories; I noted that the authors also acknowledged this in the discussion section.) If there might be discrepancy, how large would it be?

Page 5: It would be helpful if the authors could provide the questionnaire or a list of questions in the supplementary.

Page 6: “Table 1 outlines the rating for each building, as well as ratings based on residents’ units, buildings’ interior, and exterior of the buildings for all the AIRP concepts. Table 2 outlines the overall rating for each building, per concept.” – should be Table 2 and Table 3.

Page 9: “The authors of the paper recognize that most OPEH residing in THP have lived in and experienced different types of physical environments.” – good point.

Page 10: “Features that support safety and security included natural lighting in communal spaces, accessible signage and information as well as having security functions throughout the building.” And “Safety and security measures can be improved through outdoor lighting, hand railings, outdoor security systems, and covered seating areas.” Will adding smart sensors for, e.g., fall protection be possible for aging in place for this GHS-supported temporary housing service (https://doi.org/10.1016/j.jamda.2013.02.018; https://doi.org/10.1109/MSP.2018.2846286)?

Page 11: “The findings from the AIRP-ENV tool can support further identification of areas for environmental interventions at housing sites targeted for OPEH.” – How does the temporary housing program in the study area compare with those in other cities or other countries in general?

Author Response

Reviewer 1:

Page 2: “Several research and innovative policy initiatives tackling home-lessness are piloted across Canada, but to date little attention is directed towards OPEH and their specific challenges” – It would be helpful to add some more details of the temporary housing program, especially regarding the studied communities. For example, are older people experiencing homelessness still considered “homeless”?
Reply: To clarify, the communities that participate in the THP are at-risk of homelessness, housing precarious, or are experiencing some form of homelessness, whether that is visible or hidden. The Canadian Observatory on Homelessness defines homeless as “the situation of an individual, family or community without stable, safe, permanent, appropriate housing, or the immediate prospect, means and ability of acquiring it.” (Gaetz, Barr, Friesen, Harris, Hill, Kovacs-Burns, Pauly, Pearce, Turner, & Marsolais, 2012). Therefore, this definition does include those who are precariously or temporarily housed. Those who are involved in the THP are no longer homeless as they are residing in temporary housing, which provides a pathway to permanent housing, preventing homelessness. Even though the THP is a temporary accommodation, individuals are in the program until stable and secure housing is obtained. These individuals in the THP carry the lived experiences and traumas that come with being homeless, previously homeless or at-risk of homelessness. The definition of OPEH has been revised to “older persons with experiences of homelessness” (See page 1). In addition, a more detailed explanation of the temporary housing program has been added to the “Study Objective” section (See page 3).

Page 2: “Lawton and Nahemow argued that individuals in their later life are especially more affected by the P-E dynamic (Wahl et al., 2012).” – It should be mentioned that the later life conditions are also partly determined by the environment that individuals lived in across early life stages, as reviewed in a recent article: https://doi.org/10.1038/s43587-021-00140-5.
Reply: Thank you for the literature suggestion, which has now been incorporated into the manuscript (See page 3).

Page 3: “Emergent research in Environmental Gerontology is demonstrating that the built environment, including transportation, outdoor areas, and housing, has also been linked to positive mental health outcomes and improved social connectivity (Black & Jester, 2020).” Whether positive or negative depends on the characteristics of the built environment. I suggest the authors remove “positive” and “improved”.
Reply: Thank you for the suggestion. The wording has been changes to align with the recommendation (See page 3).

Page 4: “These audits were done while there were still COVID-19 Pandemic restrictions in multi-unit housing” – given the conditions during the study, will the pandemic have an impact on the results? (Especially the safety and security and social activity categories; I noted that the authors also acknowledged this in the discussion section.) If there might be discrepancy, how large would it be?
Reply: As noted in the Discussion, COVID-19 may have impacted the results of this study as all community spaces within multi-unit housing were closed to residents impacting their use of common spaces. Beyond this, it is challenging to estimate other potential discrepancies as this study only collected audit data.

Page 5: It would be helpful if the authors could provide the questionnaire or a list of questions in the supplementary.
Reply: The AIRP-ENV Audit Tool has now been attached in the supplementary (Please see attachment “2021.03.11_AIRP-ENV Checklist.pdf”).

Page 6: “Table 1 outlines the rating for each building, as well as ratings based on residents’ units, buildings’ interior, and exterior of the buildings for all the AIRP concepts. Table 2 outlines the overall rating for each building, per concept.” – should be Table 2 and Table 3.
Reply: We have now clarified and updated Table 2 and 3 (please see page 8 of manuscript).

Page 9: “The authors of the paper recognize that most OPEH residing in THP have lived in and experienced different types of physical environments.” – good point.
Reply: Thank you for this acknowledgment.

Page 10: “Features that support safety and security included natural lighting in communal spaces, accessible signage and information as well as having security functions throughout the building.” And “Safety and security measures can be improved through outdoor lighting, hand railings, outdoor security systems, and covered seating areas.” Will adding smart sensors for, e.g., fall protection be possible for aging in place for this GHS-supported temporary housing service (https://doi.org/10.1016/j.jamda.2013.02.018; https://doi.org/10.1109/MSP.2018.2846286)?
Reply: Even though the smart sensors seem to be a useful technology and may contribute to aging in the right place for older adults, it may not be an appropriate suggestion for the THP or OPEH. Community-based organizations like the one that offers the THP have limited funding and staff capacity. The article suggested above by Rantz et al. (2013) highlights the importance of community-based care-coordination with the support of nurses, other health care providers, and family members. This would be a large undertaking for a non-profit, community-based organization such as the GHS. In addition to the challenges associated with implementing smart sensors in affordable housing, there are privacy concerns as well as challenges surrounding the maintenance and sustainability of an effort such as this.

Page 11: “The findings from the AIRP-ENV tool can support further identification of areas for environmental interventions at housing sites targeted for OPEH.” – How does the temporary housing program in the study area compare with those in other cities or other countries in general?
Reply: This temporary housing program (THP) has been identified as a unique program and we are unaware of comparable others. However, in our Discussion, we offer recommendations that housing designers and planners could take into consideration as they develop new programs.

Reviewer 2 Report

The research experience "Aging in the Right Place for Older Adults Experiencing Housing Insecurity: An Environmental Assessment of Temporary Housing Program" is relevant and crucial for the contemporary society. Issues like functionality, accessibility, and autonomy and identity, whereas, satisfactory or poor for safety and security, and social activity are for sure key topics for shaping indoor&outdoor built environment.

Author Response

Review 2:

No comments were made to address.

Reviewer 3 Report

1.For a clear understanding, the author may need to add some information about "how the building environment affects the homeless elderly" in the paper. There is little previous research and case support in this paper. Thus, I recommend briefly including some of this evidence in the background.

2.The data in the first sentence of the result (page 6) is inconsistent with the data described in Table 2 and Table 3 (e.g., the ratings of all dimensions of the four buildings mentioned in the article are "OK" or "Good", but the social activity of Meadow Gardens in Table 3 is "Fair"). A detailed description and some discussions of this phenomenon are required.

3.There is a problem in the presentation of the results (Lines 4 and 5, Exterior of the Buildings), which is inconsistent with the data in Table 2. The level of Hallow Place is "Fair", while the level of Arbutus Plaza and Sunset Place should be "Good".

4.I recommend adding some charts related to the interview results, such as evaluation feedback, differences between groups in different building environments, etc. in the results section. The narrative section needs the support of objective data, not only subjective description.

Author Response

Reviewer 3:

1.For a clear understanding, the author may need to add some information about "how the building environment affects the homeless elderly" in the paper. There is little previous research and case support in this paper. Thus, I recommend briefly including some of this evidence in the background.
Reply: We have revised the background section to better ground the purpose of this paper in prior research and highlighting the impacts housing stability has on OPEH. For example, on page 2 we now state, “This instability can negatively impact the health and well-being of these older adults; research shows that OPEH appear to have mental and physical health characteristics that are similar to non-homeless individuals who are 10 years older than they are (Grenier et al., 2016). Additionally, OPEH have higher rates of early mortality and spend more time in shelters than younger homeless people (Grenier et al., 2016).” In addition, on page 3 we have included information from Wang et al. (2021) highlighting the importance of one’s environment throughout the life course, which does not directly include OPEH. Of note, there is research that focuses on understanding AIRP among the general population of older adults (doi:10.1080/08882746.20 17.1398450; doi:10.1093/geront/gnz112; doi:10.1108/WWOP-08-2016-0020); however, considerations of AIRP that are relevant to OPEH are largely absent from the literature (See page 1). Furthermore, research on homelessness does not focus on the built environment, which is a main reason we undertook this environmental audit study (See page 3). We showcase the AIRP conceptual framework developed by Canham et al. (2022), which directly points to the importance of the built and natural environment of one’s housing unit and surrounding neighbourhood as a key category within the framework (See page 3). Canham et al. (2022) draw from person-environment fit theory as well as the concept of aging in place to bring the environment to the foreground of homelessness and aging research, by stating that the characteristics, design, and accessibility of the built environment of housing and the surrounding neighborhood impact an individual’s ability to AIRP (See page 3). In the AIRP conceptual paper by Canham et al. (2022) page 3 highlights supportive features of the built environment for OPEH to AIRP. These elements include “the presence of universal design features for barrier-free and accessible features (e.g., sidewalks and ramps), appropriate heating and lighting in communal and private spaces, wayfinding and navigational indicators, and homelike decor/furnishings”, which have been mentioned throughout the finding and discussion sections of the manuscript (Canham et al., 2022, p. 3).

2.The data in the first sentence of the result (page 6) is inconsistent with the data described in Table 2 and Table 3 (e.g., the ratings of all dimensions of the four buildings mentioned in the article are "OK" or "Good", but the social activity of Meadow Gardens in Table 3 is "Fair"). A detailed description and some discussions of this phenomenon are required.
Reply: We revised the first sentence to reflect the data described in Table 3 as it presents the overall scoring of each AIRP concept, as well as the description of each table (See page 6).

3.There is a problem in the presentation of the results (Lines 4 and 5, Exterior of the Buildings), which is inconsistent with the data in Table 2. The level of Hallow Place is "Fair", while the level of Arbutus Plaza and Sunset Place should be "Good".
Reply: The presentation of results have been revised in all of the sections (resident unit, interior, and exterior) to ensure accurate representation of data in the tables.

4.I recommend adding some charts related to the interview results, such as evaluation feedback, differences between groups in different building environments, etc. in the results section. The narrative section needs the support of objective data, not only subjective description.
Reply: There were no interviews collected as part of this study. We have clarified this throughout the manuscript. In addition, we note in our Limitations that the voices and lived experiences of OPEH were not captured in this larger study.

Round 2

Reviewer 1 Report

The manuscript has been substantially improved in this round of revision. I have no additional comments.